# Dynamic Analysis of microRNAs from Different Life Stages of *Rhipicephalus microplus* (Acari: Ixodidae) by High-Throughput Sequencing

**DOI:** 10.3390/pathogens11101148

**Published:** 2022-10-04

**Authors:** Jin Luo, Shuaiyang Zhao, Qiaoyun Ren, Qilin Wang, Zeyu Chen, Jingjing Cui, Yujiao Jing, Peiwen Liu, Ruofeng Yan, Xiaokai Song, Guangyuan Liu, Xiangrui Li

**Affiliations:** 1MOE Joint International Research Laboratory of Animal Health and Food Safety, College of Veterinary Medicine, Nanjing Agricultural University, Nanjing 210095, China; 2State Key Laboratory of Veterinary Etiological Biology, Key Laboratory of Veterinary Parasitology of Gansu Province, Lanzhou Veterinary Research Institute, Chinese Academy of Agricultural Science, Xujiaping 1, Lanzhou 730046, China

**Keywords:** dynamic analysis, MicroRNA (miRNA), *Rhipicephalus microplus*, high-throughput sequencing

## Abstract

MicroRNAs (miRNAs), which are small, noncoding RNA molecules, play an important regulatory role in gene expression at the posttranscriptional level. Relatively limited knowledge exists on miRNAs in *Rhipicephalus microplus* ticks in China; however, understanding the physiology of miRNA functions and expression at different developmental stages is important. In this study, three small RNA libraries were constructed for *R**. microplus* eggs, larvae, and female adults; miRNAs were detected during these developmental stages by high-throughput sequencing, with 18,162,337, 8,090,736, and 11,807,326 clean reads, respectively. A total of 5132 known miRNAs and 31 novel miRNAs were identified. A total of 1736 differentially expressed miRNAs were significantly different at a *p*-value of <0.01; in female adults, 467 microRNAs were upregulated and 376 miRNAs downregulated compared to larval tick controls. Using larvae as controls, 218 upregulated and 203 downregulated miRNAs were detected in eggs; in eggs, 108 miRNAs were upregulated and 364 downregulated compared to female adults controls. To verify the reliability of the sequencing data, RT–qPCR was applied to compare expression levels of novel miRNAs. Some differentially expressed miRNAs are involved in developmental physiology, signal transduction, and cell-extracellular communications based on GO annotation and KEGG pathway analyses. Here, we provide a dynamic analysis of miRNAs in *R**. microplus* and their potential targets, which has significance for understanding the biology of ticks and lays the foundation for improved understanding of miRNA functioning in the regulation of *R**. microplus* development. These results can assist future miRNA studies in other tick species that have great significance for human and animal health.

## 1. Introduction

Ticks are important haematophagous arthropods that act as vectors of animal and human diseases. Ticks use the blood of vertebrate animals as their only food source, and tick bites can not only cause mechanical damage but also transmit various pathogens, including *Theileria*, *Babesia*, *Rickettsia*, and *Anaplasma* [1,2,3,4,5]. Ticks are distributed worldwide, and many are parasitic on cattle, sheep and other hosts [6,7]. Cattle are the main hosts of *Rhipicephalus microplus*, a tick species that spreads a variety of pathogens, causing serious harm to the cattle industry. Unlike in other ticks, these ticks can complete their full lifecycle on cattle (one-host tick). This phenomenon is important for studying the developmental characteristics of these ticks and providing tick control.

MicroRNAs (miRNAs) are small, noncoding RNA molecules with a length of only 18–22 nt. miRNA precursors have a special hairpin structure, forming multiple bubble-like structures. Affected by enzyme digestion, the 5′ and 3′ ends of miRNA are mainly uracil (U), and the middle of the sequence is mainly adenine (A). Mature miRNAs show high similarity in different species, especially with regard to the base at positions 2–8, which is absolutely conserved in the same miRNA molecule [8]. This region is called the “seed region”, which can be divided into the (a) region and (b) region. A difference of 1–2 bases is allowed in the (a) region, whereas 2–3 base differences are allowed in “B”. The conservation of this sequence is an important link between miRNA and target gene recognition, comprising an important embodiment of its function. miRNAs are mainly involved in the regulation of mRNA at the posttranscriptional level, inhibiting or degrading target genes. Studies have shown that miRNAs widely exist in animals and plants [9,10,11,12,13]; their functional diversity manifests in their participation in cell differentiation, cell proliferation, molecular metabolism, signal transduction, inflammatory response, and antigenic effects in arthropods, including ticks, *Drosophila* and silkworms [14,15,16]. In view of the diversity of miRNA functions and their important role in biology, an increasing number of studies have been carried out on the function of miRNAs in recent years, the regulatory mechanisms of many functional genes have been revealed.

High-throughput sequencing (Hiseq) has outstanding advantages for high accuracy, high throughput, high sensitivity, and a low signal-to-noise ratio. This technology has allowed for identification of small RNA digital expression in different organisms via synthesis and sequencing [17]. Overall, Hiseq is a kind of sequencing technology based on a single molecular cluster.

The aim of this study was to analyze the expression characteristics of miRNAs during the development of *R. microplus* by high-throughput sequencing (HiSeq) and comparative genomics and to identify new miRNAs. This study provides data supporting future studies on the function of miRNAs in tick development and enriches the amount of data on tick miRNAs.

## 2. Results

### 2.1. Small RNA Library Construction and Solexa Sequencing

To identify the expression features of miRNAs in various developmental stages of *R. microplus* physiology, three small RNA libraries were constructed for eggs, larvae, and female adults and sequenced using Illumina/Solexa-HiSeq; a total of 18,405,692, 8,826,989, and 11,909,507 raw reads were obtained, respectively. Low-quality reads, adaptors, and insufficient tags were removed, with 18,162,337, 8,090,736, and 11,807,326 clean reads being retained, respectively. Of these, 1,435,283 egg sequences, 1,903,019 larval sequences, and 1,750,974 female adult sequences accounted for 7.90%, 23.52%, and 16.47% of the total reads, respectively. These reads mapped perfectly to the *Ixodes scapularis* genome (Accession: PRJNA34667). A total of 4,908,986, 1,506,602, and 1,735,790 unique sequences for eggs, larvae, and female adults, respectively, in various developmental stages remained for further analysis. Common sequences between larvae and eggs were calculated to cover 1.99% of the total; 21.96% are specific to larvae and 76.05% to eggs (Figure 1a). Between the larval and female adult stages, common sequences represent 2.08%; specific sequences comprise 65.21% and 32.71%, respectively (Figure 1b). Between female adults and eggs, 10.03% are common sequences, with 30.95% specific sequences in female adults and 59.02% in eggs (Figure 1c). Overall, the sequences common to all developmental stages represent 9.67% (Figure 1d). Therefore, many miRNAs are common, though some miRNAs are unique at different developmental stages. For example, miR-4175-3p was detected not only in eggs (copy number n = 153,456) but also in female adults (n = 55,350). miR-4419b was also commonly expressed between eggs and female adult ticks; the copy number was 107,419 in eggs and female adult ticks (n = 43). miR-5187-5p is common in both eggs (n = 9) and larvae (n = 12,738). miR-2371 was found to be expressed by both larvae and female adults, with expression levels of 22,276 and 915, respectively. miR-2126 was also identified in the two samples. For analysis of specific miRNAs, miR-483-3p (n = 30) and miR-1421ac* (n = 23) were only expressed in eggs; miR-4908-3p (n = 162), miR-669a-3-3p (n = 1408), and miR-1183 (n = 42,730) were specifically expressed in larvae and miR-516-3p (n = 42,730) and miR-26b-5p (n = 10,186) in female adults.

The length distribution of the obtained small RNAs was analyzed across libraries (Figure 2). The results showed the majority of lengths to be 18–30 nt, and there were two peaks after digestion with the Dicer and Drosha enzymes. The miRNA length range is 21–22 nt; that for siRNA is 24 nt, and that for piRNA is 30 nt. All sequence reads were annotated and classified using the GenBank and Rfam databases. The results showed that 18,162,337, 8,090,736, and 11,807,326 reads could be annotated and classified for the egg, larval, and female adult libraries, respectively. A total of 15,137,703 (4,743,445 unique sRNAs, accounting for 83.35% of total egg reads), 5,255,640 (1,395,235 unique sRNAs, accounting for 64.96% of total larval reads), and 9,426,350 (2,819,174 unique sRNAs, accounting for 79.83% of total female adult reads) reads are unannotated in the libraries, and these data were further analyzed for novel miRNAs. The classification annotation is shown in Table 1.

### 2.2. Known Conserved MicroRNAs and Differential Expression at Different Developmental Stages


Unfortunately, there are only 49 mature miRNAs in the miRBase 18.0 (http://www.mirbase.org/; accessed on 1 July 2021) database for the hard tick lineage [18,19], and these known miRNAs were used to identify additional miRNAs from the sequencing data of small RNAs using the repository of all animal miRNAs in the database. These unique small RNAs from eggs, larvae, and female adults of *R. microplus* (Table 1) were analyzed using the genome of *R. microplus* in the miRBase database. The results showed 40 known miRNAs in the egg stage, 44 in the larval tick stage, and 43 in the female adult stage (Table 2). To identify the type and number of miRNAs in the sequencing data, we mapped the clean reads to all species, and differential expression of miRNAs was analyzed between samples. The results showed that a total of 1736 microRNAs were significantly differentially expressed at a *p*–value of <0.01 during different developmental stages. When larvae were used as the control, 467 miRNAs were upregulated and 376 miRNAs significantly downregulated in female adults. In addition, 218 upregulated and 203 downregulated miRNAs were identified in eggs. Similarly, in eggs, 108 miRNAs were upregulated and 364 downregulated when compared to female adult controls (Figure 3; Appendix A). Across the different libraries, some miRNA expression levels were greater than 100,000 copies; for example, miR-4175-3p constituted 17.78% of the total clean reads. In addition, miR-184, let-7-5p, miR-4486, miR-1, miR-84a and Bantam were the most abundant in these samples (Appendix A).

### 2.3. Identification of Novel microRNA Candidates

These unannotated small molecular sequences were used for the prediction of novel miRNAs by Mireap: http://sourceforge.net/projects/mireap/, accessed on 1 July 2021 [20]. The results showed 22, 12, and 13 potential novel miRNAs in eggs, larvae, and female adult ticks, respectively. These molecules have a common feature: a stem–loop structure and free energy ranging from −43.5 kcal mol^−1^ to −19.4 kcal mol^−1^ (Appendix A).

### 2.4. Validation of MicroRNA Expression by Quantitative RT–PCR

In this study, relative RT–qPCR was applied to assess expression levels of the novel miRNAs. The results showed m0001 and m0004 to be differentially expressed in larvae and female adults (Figure 4a), and m0005 was differentially expressed in larvae and eggs (Figure 4b). Furthermore, m0014 was differentially expressed in female adults and eggs (Figure 4c), and m0006, m0007, and m0018 were differentially expressed in all developmental stages (Figure 4d). The expression levels of all novel miRNAs in eggs were validated by RT–qPCR, and the expression pattern was consistent with the Solexa sequencing results (Appendix A). Only the results for m0003 (Novel 3), m0012 (Novel 12), m0014 (Novel 14), m0025 (Novel 25), m0027 (Novel 27), and m0028 (Novel 28) were inconsistent with the Solexa sequencing results, but the trend between the HiSeq and RT–qPCR results was the same (Figure 5).

### 2.5. MicroRNA Target Gene Prediction and GO Enrichment and KEGG Pathway Analyses

Gene Ontology (GO) is a standard method for gene annotations, involving cellular component, molecular functions and biological processes, in various species [21]. Based on GO enrichment, 2189 genes were enriched in cellular components, and 57.50% of the target genes are associated with organelles, including extracellular region, macromolecular complex, and membrane-enclosed lumen. Enrichment analysis showed that 3228 genes could be assigned to different functions, with 2187 (67.80%) of the functions being related to catalytic activity, for example, PROK-G, ribozyme, and ester hydrolase. In the biological processes category, 37% of genes are involved in macromolecule metabolism, and 23% are involved in nitrogen compound metabolic processes, for a total of 2747 genes, including vesicle transport protein, ecdysone, and heat shock protein. The GO items of the targets were also subjected to enrichment analysis (Figure 6), and some specific miRNA clusters in the molecular function and biological process categories were observed. Auxiliary transport proteins occupied a significantly higher percentage in larvae under molecular function but were not identified in eggs and female adults (Figure 6, annotated with “1”). Conversely, the electron carrier and translation regulator protein showed a lower percentage in larvae than in eggs or adults. These data are provided in Figure 6 and annotated with “2” and “3”.

In addition, the target genes of novel miRNAs were predicted by using MireapV0.2 software, and gene functions were annotated by Kyoto Encyclopedia of Genes and Genomes (KEGG) as putative target genes across development stages. According to the results, a total of 4123 target genes were annotated with 268 biological processes, and most of the target genes are involved in metabolic pathways, disease occurrence, and substance metabolism. Most of the 972 (23.58%) target genes are enriched in metabolic pathways, followed by steroid hormone biosynthesis (4.29%), drug metabolism (3.74%), and exogenous substance metabolism (3.71%).

## 3. Discussion

*Rhipicephalus microplus* is a widely distributed species that not only causes mechanical damage but also harms the cattle industry. These ticks can complete their full lifecycle on cattle (one-host tick), which is very important for their rapid maturation and reproduction and even the ability to quickly spread pathogens.

MicroRNAs are considered important regulators of posttranscriptional expression of genes and have different functions in organisms, such as growth, material metabolism, cell differentiation, and disease generation [22,23,24]. Previous studies on miRNAs have focused on identifying expression of known miRNAs in plants and animals using traditional PCR and bioinformatics methods [25,26]. However, these methods are difficult to implement for the identification of novel miRNAs. HiSeq provides a digitalization analysis platform that allows high-throughput, accurate acquisition of known and novel miRNA sequences in detected samples [27]. The method can produce millions of miRNA molecules at one time and can identify the expression pattern of small RNA molecules under certain conditions. In general, HiSeq is a powerful tool for functional analysis of miRNAs [28,29,30,31]. RT–qPCR, northern blotting and microarray have been used to verify miRNA prediction results [32,33,34].

In this study, small RNA reads were obtained using HiSeq. More than 89% of the reads were 18~24 nt in length, which is consistent with the typical size of salivary gland miRNAs from *Haemaphysalis longicornis* in previous studies [13]. Although the length distribution was similar for small RNAs at different developmental stages, there were some differences in length at different sites. The results of length analysis can be used as an important basis for determining the reliability of sequencing data and can indirectly reflect the characteristics of the distribution of different kinds of small RNAs. These results provide an important reference for data analysis and useful information for the physiological functions of ticks analyzed based on the spatiotemporal expression patterns of small RNAs [8]. In our study, miRNAs accounted for a large proportion of the 18,162,337, 8,090,736, and 11,807,326 sequences acquired from different developmental stages, indicating that miRNAs have an important function in regulating the posttranscriptional expression level of target genes.

The results of analysis of miRNA expression patterns in different developmental stages of ticks showed that miRNAs are an important stage-specific regulatory factor. Different molecules may act in different stages and be directly related to specific physiological processes of ticks at specific stages, such as moulting, maintaining homeostasis, physiological development, and material metabolism. miR-1-3p was most abundant, with 466,287 read counts in various developmental stages based on conserved and differentially expressed gene analysis, and it has been reported that miR-1 plays an important role in muscular development [35,36]. In addition, Bantam, miR-4175-3p, miR-184, let-7-5p, miR-4486, and miR-84a showed high read counts, with more than 100,000 reads. Bantam is an important miRNA that regulates the growth of dendrites in sensory neurons in the epithelial cells of *Drosophila* [37]. Bantam microRNA has also been directly linked to the Hippo signalling pathway, which regulates cell proliferation and survival [38]. Although these results were obtained in *Drosophila*, recent studies have shown that human miRNAs are also related to this pathway [39]. Homothorax (Hth) and Teashirt (Tsh) are expressed in the eyes of *Drosophila* and are DNA-binding transcription factors. A previous study showed that these proteins are important for cell survival in *Drosophila* and participate in the regulation of Bantam miRNA function [38]. Overall, miRNAs play an important role in the central pacemaker of the *Drosophila* circadian rhythm; it is also possible that Bantam regulates the clock component clk, making it an important factor for regulating *Drosophila* behavior [40]. In the present study, fewer than 100 expressed reads for 1333 miRNAs were observed across different developmental stages. There were 13 novel miRNAs solely present in eggs, one novel miRNA that was identified only in larvae, and five that were only detected in female adults. Additionally, three novel miRNAs were present in every developmental stage (Table 3). These differentially expressed miRNAs play an important role in specific developmental physiological regulation.

Based on the above research, mirsan, mirseker, and findmirna software have been found to have more advantages than traditional methods when detecting known miRNAs and discovering new miRNAs [41]. Due to the hairpin structure, free energy differences may also exist in the precursor sequences of nonmiRNAs, which can create challenges for the prediction of new miRNAs [42]. The Mireap software uses complete miRNA parameter characteristics and calculation methods and can be used to predict new miRNAs [43]. Mfold and MiPred software were used to analyze stem–loop structures and obtain novel miRNAs. In this study, some miRNAs were randomly selected for validation using RT–qPCR. Among the 18 candidates, 12 miRNAs were validated, whereas six novel miRNA candidates need further experimental verification, possibly because of very low expression or false-positive results (Figure 5 and Table 3).

miRNAs regulate protein expression or degrade target mRNAs via incomplete complementary binding to 3′-UTRs [44,45]. Currently, prediction of miRNA target genes is not accurate, even though several bioinformatic tools, such as miRanda, RNAhybrid, and PicTar, have been developed; however, these software programs may have high false-positive rates [46]. There is also no available 3′-UTR database for target gene prediction of miRNAs from ticks, which has created a challenge for further understanding the physiological function of miRNAs. To provide additional insight into the physiological functions of miRNAs involved in different developmental stages of ticks, target genes were predicted using the Mireap and TargetScan 6.0 software following the rules of Allen E [47] and He S [48] (Materials and Methods). We also determined the function of miRNAs and target genes based on GO annotation and KEGG pathway analyses. Based on GO and KEGG results, the putative target genes are involved in a wide range of biological processes, such as enzyme catalytic activity, material metabolism, and molecular adhesion. Regarding GO enrichment, more than 57.50% of the genes were annotated to the intracellular component ontology in the cellular component category, more than 67.80% of the genes are involved in catalytic activity in the molecular function category, and approximately 36.80% of the genes participate in macromolecule metabolic processes. KEGG analysis showed that approximately 23.58% of the genes are related to a metabolic pathway.

Although the miRNAs/novel miRNAs of nymphs were not described in this study, the expression characteristics of miRNAs in ticks can be insight through the expression trend of miRNAs in unfed larvae, adult female ticks and eggs, such as miR-184, let-7-5p, miR-4486, miR-1, miR-84a, and Bantam could the most abundant in nymphs. It is suggested that further studies can assess differences in microRNA across all life cycle stages including the nymphal stages.

## 4. Materials and Methods

### 4.1. The Experimental Analysis Workflow

The workflow was designed for dynamic analysis of microRNAs from different life stages of *R.*
*microplus* by high-throughput sequencing (Figure 7).

### 4.2. Tick Collection and RNA Extraction and Library Preparation

In this study, *R. microplus* ticks were collected from Yongjing County, Gansu Province, and maintained under laboratory conditions. These ticks were identified using morphology by the Animal Research Institute (Lanzhou Veterinary Research Institute). Approximately 0.5 mg of eggs and 1 g of larvae were used for miRNA extraction, and another 1 g of larvae was placed in a cloth bag attached to cattle for 25 ± 3 days to produce engorged adult female ticks. To analyze the expression profile of miRNAs in *R. microplus*, unfed larvae, adult female ticks and eggs were ground in liquid nitrogen to extract total RNA using TRIzol reagent (Invitrogen, Cat No. 15596-026). The concentration of RNA was 400 ng/mL (260/280 = 2.00). Some RNAs were used to construct libraries for small RNA sequencing, and some were used for miRNA validation. The synthesis of first-strand cDNAs was performed according to the protocol for transcriptase XL (Avian Myeloblastosis Virus, AMV) (TaKaRa, Shiga, Japan) with a loop primer of miRNAs and oligo dT18. PCR was performed to obtain the miRNAs sequence. The PCR product was ligated to the T-Easy vector (TaKaRa, Japan), which was transformed into JM109 (TaKaRa, Japan). The sequence was obtained by GenScript (Nanjing, China).

### 4.3. Small RNA Isolation and High-Throughput Sequencing

The quality of total RNA was analyzed using a Shimadzu 206-97213C BioSpec-nano analyzer system and by denaturing polyacrylamide gel electrophoresis. Small RNA was enriched using a miScript microRNA isolation kit (QIAGEN, Biotechnology Co., Ltd., China), and a small RNA library was generated according to Illumina sample preparation instructions [49].

### 4.4. Small RNA Bioinformatics Analysis

Raw sequences were produced by HiSeq using Illumina MiSeq equipment (BGI. Tech). These clean reads were analyzed after removing low-quality reads, reads shorter than 12 nt, and repeat and adaptor sequences. The clean reads were assembled using the SOAP software (http://soap.genomics.org.cn/soapdenovo.html:1.05; accessed on 1 July 2021), and the length distribution was summarized. To identify miRNAs from ticks, all clean reads were mapped to the genome of *Rhipicephalus microplus* (assembly ASM1333972v1), *I**xodes scapularis* (assembly ASM1692078v2), *Haemaphysalis longicornis* (assembly ASM1333976v2) by the Bowtie software (http://bowtie-bio.sourceforge.net/manual.shtml) [50], a process that ensures the uniqueness of miRNA annotation and the host miRNAs can be removed. The following principles were followed: rRNAetc (in which GenBank > Rfam) > known miRNA > repeat > exon > intron [51]. Unannotated sequences were used for prediction of novel miRNAs. The hairpin structure, digestion site of Dicer, and minimum free energy (≥20 kcal/mol or *p*-value of >0.05) are important parameters for predicting miRNAs [20,38,52,53]. Prediction of novel miRNAs was performed by MiPred (http://www.bioinf.seu.edu.Cn/mirna/; accessed on 1 July 2021).

### 4.5. Target Prediction for Novel miRNA Candidates

At present, no 3′-UTR database is available for predicting target genes of miRNAs. Therefore, a new strategy was implemented to compare miRNA sequences with those in the tick EST database in NCBI to determine hypothetical target genes of novel miRNAs [47,48]. In this study, multiple softwares (RNAhybrid and miRanda) were used to predict the target genes of miRNAs. Moreover, then selects the corresponding intersection or union as the prediction result. Furthermore, the filter according to the corresponding filtering conditions such as free energy and score value, and select the results supported by at least two target gene prediction software.

### 4.6. Differential Expression of Known miRNAs and GO Enrichment and KEGG Pathway Analyses

In this study, differentially expressed miRNAs between two samples were calculated by plotting the log2-ratio and generating a scatter plot. Normalized expression = actual miRNA count/total count of clean reads *1,000,000, and Fold_change = log2 (treatment/control).

GO provides a method for analysing biological processes, cellular components and molecular functions. First, candidate target genes were mapped to GO terms (http://www.geneontology.org/; accessed on 1 July 2021) [21]; the number of genes for each term was then calculated, and the hypergeometric method was applied to find GO terms significantly abundant among the target candidate genes compared with the genomic background. Moreover, BGI Co. uses the reference software GO:: termfinder’ (http://www.yeastgenome.org/help/analyze/go-term-finder; accessed on 1 July 2021) develop a formula for GO entries.
$$p=1-\overset{m-1}{\underset{i=0}{\sum }}\frac{\left(\begin{array}{c} M\\ i \end{array}\right)\left(\begin{array}{c} N-M\\ n-i \end{array}\right)}{\left(\begin{array}{c} N\\ n \end{array}\right)}$$

In the formula, “*N*” is the number of all genes with GO annotation; “*n*” is the number of target gene candidates; “*M*” is the number of all genes that are annotated to a certain GO term; “*m*” is the number of target gene candidates in “*M*”. The Bonferroni correction for the *p*-value was used to obtain a corrected *p*-value. For GO terms, a *p*–value of ≤0.05 was defined as significantly enriched in target genes of miRNAs.

KEGG pathway analysis can also help determine the functions of candidate target genes [54], predicting the role of protein interaction networks in various cell activities. KEGG is widely used as a reference knowledge base for integration and interpretation of large-scale data sets obtained by genome sequencing and other high-throughput experimental technologies. The calculation formula is the same as that used for GO analysis.

### 4.7. Real-Time Quantitative PCR

Stem-loop RT–qPCR was used to analyze novel miRNA expression in *R. microplus*. A fixed stem loop adaptor sequence (5′-GTC GTA TCC AGT GCA GGG TCC GAG GTA TTC GCA CTG GAT ACG AC-3′) was used to quantify miRNA expression; the reverse universal primer (10 × miScript Universal primer) was provided by QIAGEN Biotechnology Co., Ltd., China). All primers were synthesized by Shenggong Co., Ltd., China, according to the mature sequence of miRNAs (Table 3). Amplification was performed using Mx3000pTM SYBR Green RT–qPCR Analyser (QIAGEN Biotechnology Co, Ltd., China). Briefly, 2 μg of RNA was reverse transcribed using miScript Ⅱ miRNA cDNA Synthesis Kit (QIAGEN Biotechnology Co., Ltd., China). The reverse transcription reaction system included 4 μL of 5 × miScript HiFlex Buffer, 2 μL of 10 × miScript Nucleics Mix, 2 μL of miScript Reverse Transcriptase Mix and RNase-Free dH_2_O in a final volume of 20 μL. All RT–qPCR programs and reaction mixtures were prepared according to the manufacturer’s protocol [55,56]. Each sample was replicated three times. miRNA levels in samples from different developmental stages were determined individually. Each miRNA level is expressed as the 2^−^^ΔΔCT^ mean ± SE.

## 5. Conclusions

Solexa HiSeq provides a crucial tool for identification of miRNAs in various developmental stages of ticks. In this study, three small RNA libraries were constructed from tick eggs, larvae, and female adults. We evaluated the functions of miRNAs in tick development, and 17 novel miRNAs were identified, greatly enriching the information on miRNAs in ticks and providing data support for a comprehensive understanding of the possible physiological functions of miRNAs. From the novel miRNAs, 1203 significantly differentially expressed targets were predicted, and the GO and KEGG results showed that these target genes regulated by miRNAs play an important role in the development of *R. microplus*. This study provides further insight into the miRNA-mediated regulation of target genes in various developmental stages of ticks.

## Figures and Tables

**Figure 1 pathogens-11-01148-f001:**
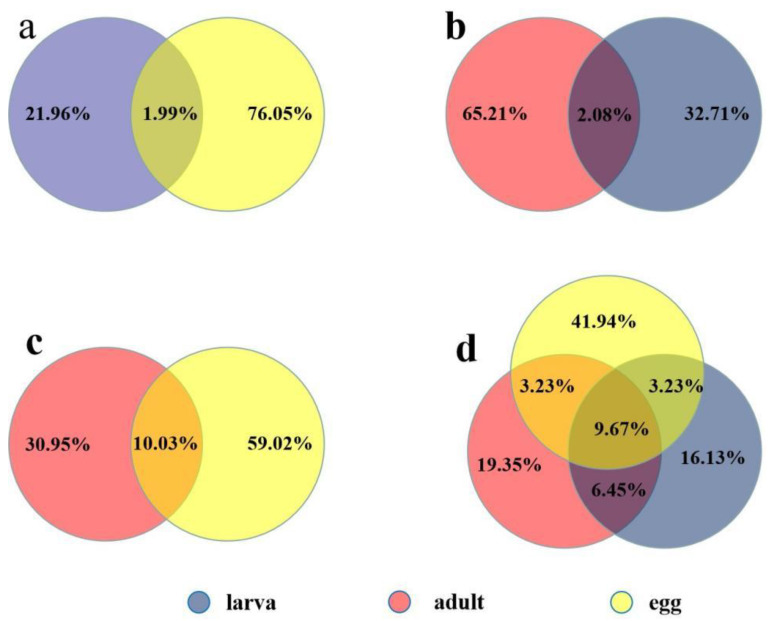
Analysis of common and unique tag sequences in various developmental stages of *Rhipicephalus microplus* by venn chart uniq_sRNAs. (**a**): The percentage of shared sRNAs between larvae and eggs; (**b**): the percentage of shared sRNAs between female adults and larvae; (**c**): the percentage of shared sRNAs between female adults and eggs; and (**d**): the percentage of shared sRNAs across all developmental stages.

**Figure 2 pathogens-11-01148-f002:**
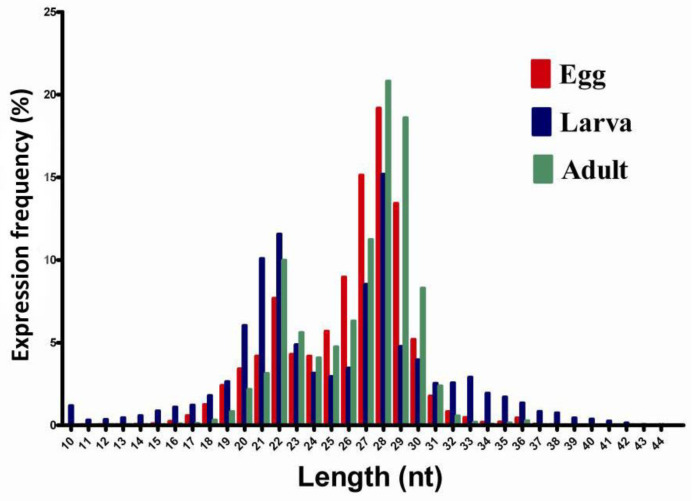
Length distribution and abundance of small RNA sequences at different developmental stages of *Rhipicephalus microplus*. The X-axis represents the base sites in the same read sequence. The Y-axis indicates the percentage of clean reads at different sites based on all small RNAs. Red indicates clean reads from eggs; blue indicates clean reads from larvae; green indicates clean reads from female adults.

**Figure 3 pathogens-11-01148-f003:**
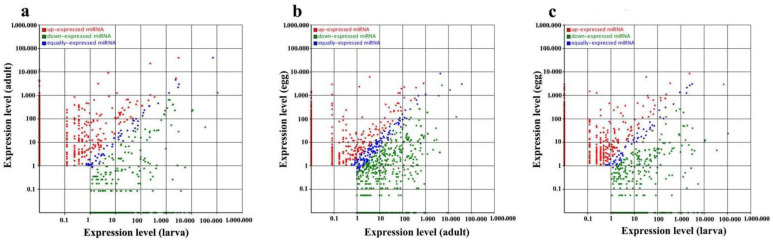
Comparison of expression levels of different miRNAs between two samples by scatter plot (control: X/treatment: Y). Panel (**a**) represents miRNAs with differential expression between female adults and larvae; (**b**) represents miRNAs with differential expression between eggs and female adults; (**c**) represents miRNAs with differential expression between eggs and larvae.

**Figure 4 pathogens-11-01148-f004:**
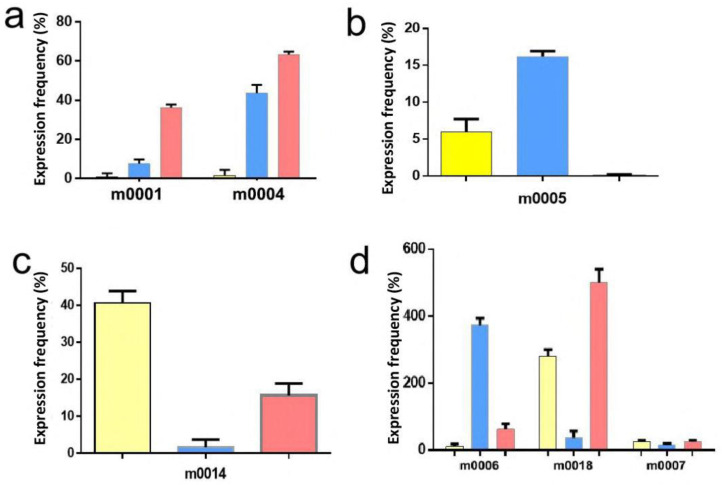
Novel miRNAs were identified by RT–qPCR. (**a**) RT–qPCR results for 2 novel miRNAs (m0001 and m0004) in larvae and female adults. (**b**) RT–qPCR results for 1 novel miRNA (m0005) in larvae and eggs. (**c**) RT–qPCR results for 1 novel miRNA (m0014) in female adults and eggs. (**d**) RT–qPCR results for 3 novel miRNAs (m0006, m0018, and m0007) in all developmental stages. Furthermore, the yellow color represents in egg, blue color represents in larva, and red color represents adult.

**Figure 5 pathogens-11-01148-f005:**
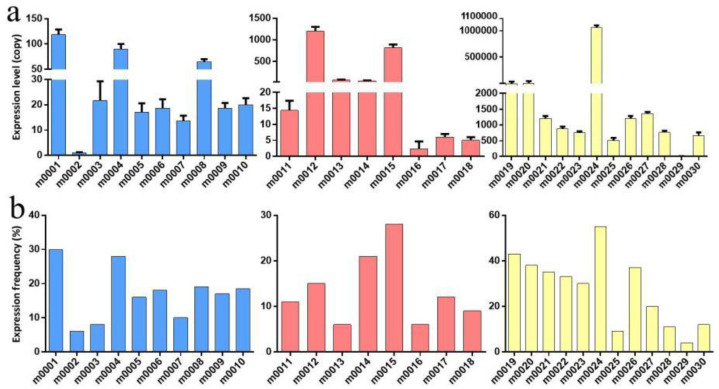
Potential novel miRNA expression across all developmental periods was confirmed by RT–qPCR. Y-axis: relative quantity (dRn) with a log-based scale; X-axis: sample name. Panel (**a**) represents the expression level across the three libraries, which was detected by RT–qPCR. Panel (**b**) represents the expression trend in larvae, eggs, and adults, which was detected using high-throughput sequencing. The blue color represents in larva, red color represents adult, and yellow color represents in egg.

**Figure 6 pathogens-11-01148-f006:**
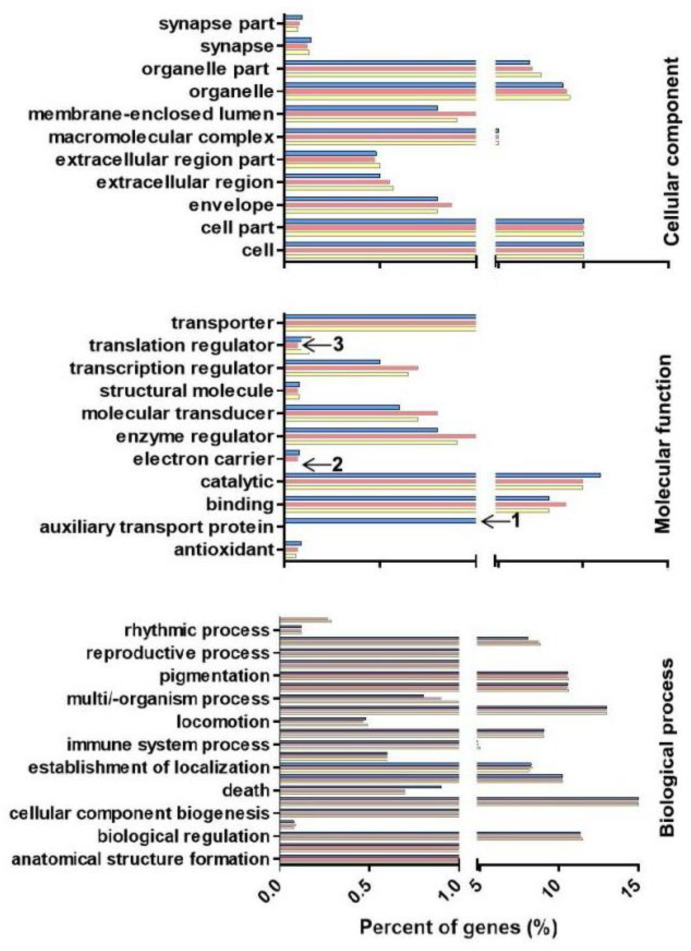
Target genes for novel-miRNAs were classified and annotated by gene2go in Gene Ontology. The blue color represents in larva, red color represents adult, and yellow color represents in egg. Furthermore, the “1” showed that auxiliary transport proteins occupied a significantly higher percentage in larvae under molecular function but were not identified in eggs and female adults. The “2” and “3” showed that a lower percentage in larvae than in eggs or female adults.

**Figure 7 pathogens-11-01148-f007:**
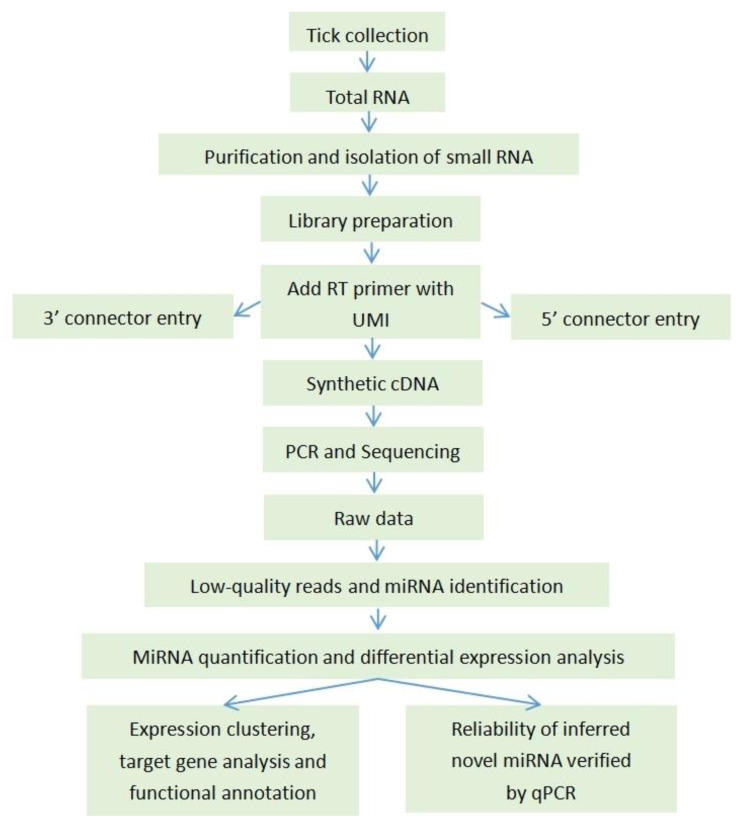
Tick-derived miRNA sequencing and analysis workflow.

**Table 1 pathogens-11-01148-t001:** Distribution of small RNAs among different categories at different developmental stages of *Rhipicephalus microplus* ticks.

Category	Egg				Larvae				Adult			
	Unique sRNAs	Percent (%)	Total sRNAs	Percent (%)	Unique sRNAs	Percent (%)	Total sRNAs	Percent (%)	Unique sRNAs	Percent (%)	Total sRNAs	Percent (%)
Total	4,908,986	100	18,162,337	100	1,506,602	100	8,090,736	100	2,913,657	100	11,807,326	100
miRNA	70,216	1.43	892,827	4.92	46,138	3.06	2,149,014	26.56	42,797	1.47	1,648,316	13.96
rRNA	64,751	1.32	1,665,040	9.17	43,714	2.90	575,299	7.11	30,408	1.04	484,924	4.11
repeat	165	0.00	285	0.00	52	0.00	92	0.00	87	0.00	231	0.00
snRNA	3713	0.08	20,631	0.11	1761	0.12	6378	0.08	2654	0.09	16,059	0.14
snoRNA	214	0.00	333	0.00	1077	0.07	1392	0.02	849	0.03	2142	0.02
tRNA	26,482	0.54	445,518	2.45	18,625	1.24	102,921	1.27	17,688	0.61	229,304	1.94
unann	4,743,445	96.63	15,137,703	83.35	1,395,235	92.61	5,255,640	64.96	2,819,174	96.76	9,426,350	79.83

**Table 2 pathogens-11-01148-t002:** Alignment of known miRNAs of *Rhipicephalus microplus* from experimental data and miRbase databases.

	miRNA	Unique sRNAs Matched to miRNA Precursors	Total sRNAs Matched to miRNA Precursors
**Known miRNA in miRBase**	49	——	——
Egg	40	405	78,860
Larvae	44	517	1,439,717
Adult	43	523	746,431

**Table 3 pathogens-11-01148-t003:** Statistics for all novel miRNAs at different developmental stages of *Rhipicephalus microplus* and summary of miRNA primers used for RT–qPCR.

Novel miRNA	Mature Sequences (5′-3′)	Primer Sequences	Development Stages	Novel miRNA	Mature Sequences (5′-3′)	Primer Sequences	Development Stages
m0001	ACTCGAGCTGCCCGTGCAAAAC	ACTCGAGCTGCCCGTG	Larva, adult	m0017	TAAGTTAATCTCCAAGCCCAAT	TAAGTTAATCTCCAAG	adult
m0002	ATCATAAGGATATCATCAATATT	ATCATAAGGATATCAT	Larva	m0018	CTGGTTTTCACAATGATCGTCC	CTGGTTTTCACAATGA	All stages
m0003	TATACGTCCAAAGCACTGAGG	TATACGTCCAAAGCAC	Larva	m0019	ACGTGCTGCATCAGGTGCTTGTGA	ACGTGCTGCATCAGGT	egg
m0004	CCTCACTCAGTTTGGCTGTGG	CCTCACTCAGTTTGGC	Larva, adult	m0020	GTCCGGAAAATCGGTCGGCGA	GTCCGGAAAATCGGTC	egg
m0005	TTTCATGTGACTTTTGAGGGC	TTTCATGTGACTTTTG	Larva, egg	m0021	TGAGCATGGTTTTCGGCAACT	TGAGCATGGTTTTCGG	egg
m0006	CCTTATCATTCGACTGTCCAGA	CCTTATCATTCGACTG	All stages	m0022	TGAACCAACTGAACGACTGAA	TGAACCAACTGAACGA	egg
m0007	TTGGGAAACAGAAGAGCGACGC	TTGGGAAACAGAAGAG	All stages	m0023	GAGTGAAAGTAGGACGCCCA	GAGTGAAAGTAGGACG	egg
m0008	GTGACTTCTCCGGTGCTGTGGA	GTGACTTCTCCGGTGC	Larva	m0024	TAGTGGTTAGGATACCTGGCT	TAGTGGTTAGGATACC	egg
m0009	AAAAATTGTGGTAGTGTCAAGC	AAAAATTGTGGTAGTG	Larva	m0025	TATGTGGTATCGTTACAAGTG	TATGTGGTATCGTTAC	egg
m0010	CTTCCCAAGCAGTTCCTGAAG	CTTCCCAAGCAGTTCC	Larva	m0026	GGCCCGTTGGTCTAGGGGTAT	GGCCCGTTGGTCTAGG	egg
m0011	AGGCATCTTTTGGAGTGCAAATG	AGGCATCTTTTGGAGT	adult	m0027	TGGGCTAGTTGGTATGGCAT	TGGGCTAGTTGGTATG	egg
m0012	TCGGATCCCATCCTCGTCGCCA	TCGGATCCCATCCTCG	adult	m0028	TCGGACGGCATCAAGAAACGT	TCGGACGGCATCAAGA	egg
m0013	TGGGCTTCCACGACGGCGGCAG	TGGGCTTCCACGACGG	adult	m0029	TTATTTATTTAGTACATACT	TTATTTATTTAGTACA	egg
m0014	GCGGAGCATTCGCGGTTGGCGGA	GCGGAGCATTCGCGGT	Adult, egg	m0030	TTGACAAGCAACTATGTATCA	TTGACAAGCAACTATG	egg
m0015	TCGAATCCTGTCGACTGCGCCA	TCGAATCCTGTCGACT	adult	m0031	GATGAGTGTGGATCTAGTCATG	GATGAGTGTGGATCTA	egg
m0016	TCCTGACCAAATGAGTGATGAGCA	TCCTGACCAAATGAGT	adult				

## Data Availability

The data supporting the conclusions in this study are included in the article.

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
