# Peer review of "Dynamic Analysis of microRNAs from Different Life Stages of Rhipicephalus microplus (Acari: Ixodidae) by High-Throughput Sequencing"

_pathogens, 2022, doi:10.3390/pathogens11101148_

Round 1

Reviewer 1 Report (Previous Reviewer 3)

The authors have made one critical and I am afraid fatal mistake. 

B. microplus has a feeding nymphal stage on the host between the feeding larvae and the feeding adult. If the authorssampled only off host stages then theymissed the nymphal stage. If theysampled on the host then they must have mixed in nymphs with either the larvae or the adults. When I reviewed this manuscript the first time I assumed they must have done the former and had inadequately described theirmethods and results. The fact now is that the authors were unaware that there is a nymphal stage. 

Whatever their sampling scheme was, they will have to completely redo the study or redo their analysis to account for having left out the nymphal stage.

I am aghast at this blunder. 

Author Response

Q1: The authors have made one critical and I am afraid fatal mistake. B. microplus has a feeding nymphal stage on the host between the feeding larvae and the feeding adult. If the authors sampled only off host stages then they missed the nymphal stage. If they sampled on the host then they must have mixed in nymphs with either the larvae or the adults. When I reviewed this manuscript the first time I assumed they must have done the former and had in adequately described their methods and results. The fact now is that the authors were unaware that there is a nymphal stage. 

What ever their sampling scheme was, they will have to completely redo the study or redo their analysis to account for having left out the nymphal stage.

I am aghast at this blunder. 

R: Thank you very much for the comment. Rhipicephalus microplus is one-host tick. So their full lifecycle on cattle from un-engorged larvae to engorged adult ticks and to eggs, in this process, the ticks have no obvious nymph stage. So it is difficult for collect nymphs. And about the segment for ticks collected has been added to “Materials and Methods”.

Reviewer 2 Report (Previous Reviewer 2)

The manuscript, “Dynamic analysis of microRNAs from different physiological stages of Rhipicephalus (Boophilus) microplus by high-throughput sequencing,” reports new information regarding the presence and expression of new and previously known miRNAs in eggs, larvae and adults of Rhipicephalus microplus, an important vector of cattle diseases.  As such, it is well within the scope of the journal and is worthy of publication.  The present manuscript is improved from the previous version, and although there appear to be a few slight problems with English language expression, overall, the paper is understandable in its present form.  Never-the-less, since the authors describe the length of small RNAs found, and mention siRNAs and piRNAs (lines 107-108), it would be beneficial to describe what is known regarding the characteristics and properties of these small RNAs in the introduction.  In addition, lines 287-290 state that some miRNAs were randomly selected for validation, and that 6 novel miRNA candidates need further experimental verification, but do not provide an explanation as to why the 18 miRNAs selected were randomly chosen, rather than concentrating on validation of the novel, previously unknown candidate miRNAs.  In my opinion, the authors spend an inordinate effort for KEGG and GO-term analyses of potential target genes in an effort to derive functional significance for the miRNAs.  KEGG and GO-terms are very broad functional classifications assigned to individual genes based on their known or presumptive functions, which can be useful to help elucidate the likely function of novel genes but seem to be of VERY limited value in describing potential significance of the miRNAs reported in the present paper.  This is simply my opinion and does not require revision of the manuscript.

Author Response

The manuscript, “Dynamic analysis of microRNAs from different physiological stages of Rhipicephalus (Boophilus) microplus by high-throughput sequencing,” reports new information regarding the presence and expression of new and previously known miRNAs in eggs, larvae and adults of Rhipicephalus microplus, an important vector of cattle diseases.  As such, it is well within the scope of the journal and is worthy of publication.  The present manuscript is improved from the previous version, and although there appear to be a few slight problems with English language expression, overall, the paper is understandable in its present form.  Never-the-less, since the authors describe the length of small RNAs found, and mention siRNAs and piRNAs (lines 107-108), it would be beneficial to describe what is known regarding the characteristics and properties of these small RNAs in the introduction.  In addition, lines 287-290 state that some miRNAs were randomly selected for validation, and that 6 novel miRNA candidates need further experimental verification, but do not provide an explanation as to why the 18 miRNAs selected were randomly chosen, rather than concentrating on validation of the novel, previously unknown candidate miRNAs.  In my opinion, the authors spend an inordinate effort for KEGG and GO-term analyses of potential target genes in an effort to derive functional significance for the miRNAs.  KEGG and GO-terms are very broad functional classifications assigned to individual genes based on their known or presumptive functions, which can be useful to help elucidate the likely function of novel genes but seem to be of VERY limited value in describing potential significance of the miRNAs reported in the present paper.  This is simply my opinion and does not require revision of the manuscript.

R: Thank you very much for your comments. We randomly selected some miRNAs for verification. It should be noted that these selected miRNAs are all novel miRNAs. By RT-PCR analysis, on the one hand, it is verified that these novel miRNAs are from ticks; on the other hand, the reliability of sequencing data can be verified by comparing sequencing data with RT-PCR results.

Thank you very much for your comments on KEGG and GO term. Indeed, the analysis of KEGG and GO enrichment is a complex process. However, the analysis of the function of miRNA target genes will have important reference value for us to study the regulation mechanism of miRNA in the future.

This manuscript is a resubmission of an earlier submission. The following is a list of the peer review reports and author responses from that submission.

Round 1

Reviewer 1 Report

The author’s sequenced microRNA’s from eggs, larvae and fed adult Rhipicephalus microplus. In total 5132 known and 31 novel miRNAs were identified. Each life stage presented different miRNA’s that was up- or down-regulated compared to other life stages. The methodology used and analysis performed is on standard. Potential targets for these miRNA’s were identified and include processes such as development, signal transduction, extra-cellular communications. The study is important given the novel information that was generated. It should serve as an important resource for future studies on the functions of these miRNA’s. Some issues follow below that require major revision, notably that the manuscript in current form is confusing since the workflow is not logically presented.

Line 31: Please check that font sizes are consistently the same in the text.

Line 40: Start sentences with full genus name, i.e. Rhipicephalus …. (check whole manuscript)

Line 43: Italicize all genus names. Check whole manuscript.

Line 50: They play …

Line 56: The study mentions that there is few reports from China. However, the tick used is a globally distributed tick and the results are therefore globally unique and reference to China is therefore not necessary. I would suggest that the authors remove references in the text that refer to the uniqueness of the study being that it has not been done in China before.

Line 64-80: The authors analyzed the raw reads to determine how many are shared or unique between different life stages (Figure 1). However, these reads were reduced to 5163 miRNA’s. Does comparison of the final miRNA’s yield the same percentages shared/unique between the life stages? It would be good if the authors can confirm this, since noise in reads may have been reduced in the final assembly.  As such, I am not sure whether the raw read figure or the final transcriptome should be compared.

Line 81-91: Similar to above, should the raw reads be analyzed rather than the final assembled and classified miRNA’s?

Line 96-113: The results seem to be confusing. While it seems as if the raw reads were analyzed, these are reduced to much less miRNA’s which from the materials and methods seem to derive from the assembly that only yielded a few thousand miRNA’s. I would suggest that all data would be reduced to the final miRNA dataset (5163) and analyzed from there. Total reads (line 100) and coverage (line 111) is becoming confused here.

Line 116-122: Again, raw reads are here assigned as unique unannotated sRNA’s. The large mapping of these reads to the Ixodes scapularis genome is also problematic. It is unclear whether millions of miRNA’s exist, or whether only a few thousand exist, or whether some artifacts exist due to genomic contamination that escape the pipeline. Especially once so few novel miRNA’s (tens) are eventually reported.

Line 249-273: The authors discuss here a variety of miRNA molecules previously found in other organisms and indicate that 1333 miRNA were identified with 19 in specific life stages. Most of the results focus on the novel miRNA and no analysis is presented on known miRNA. As such, the current paragraph creates the expectation that insights into the functions of known annotated miRNA’s in this tick will be presented and discussed, but the reader is then left in a sort of limbo. This should be differentiated from the analysis of the target genes (lines 287-305). What the reader may look for is answers to the question of whether ticks also possess Bantam, miR-1-3p, miR-41157-3p or any other miRNA with a known name. Can the authors perhaps provide a table with tick homologs to well known miRNA from those annotated? In this regard, is there a correlation between miRNA annotation and the targets identified?

Line 341-345: Can the authors describe the new strategy in more detail rather than just referring to references?

General: The study in current form is quite confusing since it seems to mix raw read analysis and final assembled miRNA analysis in a mixed manner. It is therefore difficult to follow and understand what the numbers mean. Can the data analysis flow be presented in a more logical manner? Why map raw reads (clean reads) to a genome rather than the final contigs? A supplementary figure that depicts that workflow may also help readers to understand the data analysis workflow and rationale behind the workflow design.

Author Response

Point 1: Line 31: Please check that font sizes are consistently the same in the text.

Response 1: Thank you for this suggestion. The font sizes have been standardized.

Point 2: Line 40: Start sentences with full genus name, i.e. Rhipicephalus …. (check whole manuscript)

Response 2: Thank you very much. This part has been revised, and the whole manuscript has also been checked.

Point 3: Line 43: Italicize all genus names. Check whole manuscript.

Response 3: Thank you very much. All genus names are italicized in the manuscript.

Point 4: Line 50: They play …

Response 4: Thank you; this error has been corrected.

Point 5: Line 56: The study mentions that there is few reports from China. However, the tick used is a globally distributed tick, and the results are therefore globally unique and reference to China is therefore not necessary. I would suggest that the authors remove references in the text that refer to the uniqueness of the study being that it has not been done in China before.

Response 5: Thank you for this suggestion. The sentence has been removed.

Point 6: Line 64-80: The authors analyzed the raw reads to determine how many are shared or unique between different life stages (Figure 1). However, these reads were reduced to 5163 miRNA’s. Does comparison of the final miRNA’s yield the same percentages shared/unique between the life stages? It would be good if the authors can confirm this, since noise in reads may have been reduced in the final assembly. As such, I am not sure whether the raw read figure or the final transcriptome should be compared.

Response 6: Thank you for this suggestion. Accordingly, we have provided the common and unique miRNA names and copy numbers between different samples.

Point 7: Line 81-91: Similar to above, should the raw reads be analyzed rather than the final assembled and classified miRNA’s?

Response 7: The raw reads were analysed.

Point 8: Line 96-113: The results seem to be confusing. While it seems as if the raw reads were analyzed, these are reduced to much less miRNA’s which from the materials and methods seem to derive from the assembly that only yielded a few thousand miRNA’s. I would suggest that all data would be reduced to the final miRNA dataset (5163) and analyzed from there. Total reads (line 100) and coverage (line 111) is becoming confused here.

Response 8: Thank you for this suggestion. The paragraph has been rewritten.

Point 9: Line 116-122: Again, raw reads are here assigned as unique unannotated sRNA’s. The large mapping of these reads to the Ixodes scapularis genome is also problematic. It is unclear whether millions of miRNA’s exist, or whether only a few thousand exist, or whether some artifacts exist due to genomic contamination that escape the pipeline. Especially once so few novel miRNA’s (tens) are eventually reported.

Response 9: Thank you for this suggestion. Unannotated sequences were used for prediction of novel miRNAs by some parameter of miRNA production in “Materials and Methods (Small RNA bioinformatics analysis)”.

Point 10: Line 249-273: The authors discuss here a variety of miRNA molecules previously found in other organisms and indicate that 1333 miRNA were identified with 19 in specific life stages. Most of the results focus on the novel miRNA and no analysis is presented on known miRNA. As such, the current paragraph creates the expectation that insights into the functions of known annotated miRNA’s in this tick will be presented and discussed, but the reader is then left in a sort of limbo. This should be differentiated from the analysis of the target genes (lines 287-305). What the reader may look for is answers to the question of whether ticks also possess Bantam, miR-1-3p, miR-41157-3p or any other miRNA with a known name. Can the authors perhaps provide a table with tick homologs to well known miRNA from those annotated? In this regard, is there a correlation between miRNA annotation and the targets identified?

Response 10: Thank you for this suggestion. Known miRNAs were analysed, and the name and expression level are also provided in Additional file 2.

Point 11: Line 341-345: Can the authors describe the new strategy in more detail rather than just referring to references?

Response 11: Thank you for this suggestion. More detailed miRNA and target gene methods have been added to this paragraph.

Point 12: General: The study in current form is quite confusing since it seems to mix raw read analysis and final assembled miRNA analysis in a mixed manner. It is therefore difficult to follow and understand what the numbers mean. Can the data analysis flow be presented in a more logical manner? Why map raw reads (clean reads) to a genome rather than the final contigs? A supplementary figure that depicts that workflow may also help readers understand the data analysis workflow and rationale behind the workflow design.

Response 12: Thank you for this suggestion. According to the reviewer’s comments, we added the workflow of the experimental analysis to better understand the systematic nature of the experiment (Figure 1); the numbers of the other figures have also been changed.

Reviewer 2 Report

The manuscript contains interesting and potentially significant new data on presumptive miRNAs expressed in the important tick, Rhipicephalus (Boophilus) microplus, however, the manuscript is in need of reorganization and rewriting.  The introduction is inadequate to introduce the topic and significance of the study, lacking definition of important terminology at first mention (i.e., sRNAs should be defined including subtypes and characteristics of each).  In addition, there is a glaring false statement in lines 46-47, "Unlike what occurs on other hosts, these ticks cannot complete their full lifecycle on cattle."  This is incorrect, R. microplus is a one-host tick, meaning that it does complete its lifecycle on a single host, as opposed to two- or three-host ticks, such as Ixodes scapularis.

Table 1 clearly contains results of the study, yet it is not mentioned in the text until line 272 (M&M) and is not referred to within the Results section at all.  Similarly, the citations are not listed in the order that they appear in the text, and the Figures and Tables do not appear either grouped together following the text, nor are they appropriately placed and separated within the text.  There is a great deal of analytical material presented without any explanation of what it means or why it is important.  The use of Ixodes scapularis genome as a reference genome is not inappropriate, but why was no attempt made to utilize the R. microplus genome as well? It further appears that the authors misuse the term "specific" in relation to tick lifestage where the presumptive miRNAs are found (as used in the text, e.g., line 75, it suggests that 76.05% of the common sequence was found only in eggs.  Table 2 is poorly formatted with a single digit carried over to the following line due to inadequate column width. Figure 1 contains Venn diagrams for "uniq_sRNAs", however the numbers for each lifestage do not add up to 100% as they should.  The labels on the axes of several Figures are much too small to read.  The sources of extracted total RNA used in sequencing (lines 311-316) are inadequately described, particularly as the authors spend a great deal of effort comparing and contrasting differences obtained between lifestages, but fail to clearly describe whether the adults were fed or unfed, age, size, and other significant details, similarly details on egg production, age & incubation temp, and post hatch age of the larvae.  In addition, full details of RNA isolation should also be provided.

Author Response

Point 1:The manuscript contains interesting and potentially significant new data on presumptive miRNAs expressed in the important tick, Rhipicephalus (Boophilus) microplus, however, the manuscript is in need of reorganization and rewriting.  The introduction is inadequate to introduce the topic and significance of the study, lacking definition of important terminology at first mention (i.e., sRNAs should be defined including subtypes and characteristics of each).  In addition, there is a glaring false statement in lines 46-47, "Unlike what occurs on other hosts, these ticks cannot complete their full lifecycle on cattle."  This is incorrect, R. microplus is a one-host tick, meaning that it does complete its lifecycle on a single host, as opposed to two- or three-host ticks, such as Ixodes scapularis.

Response 1: Thank you very much for the suggestion. The introduction section has been rewritten. The definition of miRNA has been supplemented. The life history of Rhipicephalus microplus is now described as follows: Cattle are the main hosts of Rhipicephalus (Boophilus) microplus, a tick species that spreads a variety of pathogens, causing serious harm to the cattle industry. Unlike in other ticks, these ticks can complete their full lifecycle on cattle (one-host tick).

Point 2: Table 1 clearly contains results of the study, yet it is not mentioned in the text until line 272 (M&M) and is not referred to within the Results section at all. Similarly, the citations are not listed in the order that they appear in the text, and the Figures and Tables do not appear either grouped together following the text, nor are they appropriately placed and separated within the text.  There is a great deal of analytical material presented without any explanation of what it means or why it is important. The use of Ixodes scapularis genome as a reference genome is not inappropriate, but why was no attempt made to utilize the R. microplus genome as well? It further appears that the authors misuse the term "specific" in relation to tick lifestage where the presumptive miRNAs are found (as used in the text, e.g., line 75, it suggests that 76.05% of the common sequence was found only in eggs.  Table 2 is poorly formatted with a single digit carried over to the following line due to inadequate column width. Figure 1 contains Venn diagrams for "uniq_sRNAs", however the numbers for each lifestage do not add up to 100% as they should.  The labels on the axes of several Figures are much too small to read.  The sources of extracted total RNA used in sequencing (lines 311-316) are inadequately described, particularly as the authors spend a great deal of effort comparing and contrasting differences obtained between lifestages, but fail to clearly describe whether the adults were fed or unfed, age, size, and other significant details, similarly details on egg production, age & incubation temp, and post hatch age of the larvae.  In addition, full details of RNA isolation should also be provided.

Response 2: Thank you very much for the suggestion. All figures and tables have been arranged in the correct order. Some sentences were reorganized and discussed. Regarding the reference genome, the R. microplus genome has not yet been published, and only the genome of I. scapularis is available in the miRBase database. However, in the process of data analysis, we compared the genomes of R. microplus and I. scapularis and noted high similarity between them. Therefore, the reference genome is still valuable to the sequencing data. Moreover, through genome comparisons between different species, we can better understand the conservation and broad spectrum of miRNAs in different tick species, which is more conducive to the study of miRNAs in tick function.

Line 75, the word "specific" is replaced by "exist in ---".

The width of all tables has been adjusted.

Figure 1. The number of each life stage in the Wien diagram containing "uniq_srnas" does not add up to 100%. This is mainly because the types of small RNAs include miRNAs, siRNAs, piRNAs, etc. Therefore, the total root content is not 100%.

In addition, full details of RNA isolation should also be provided.

Reviewer 3 Report

The study provides statistics on the numbers, frequency, activity and function of miRNA’s in the developmental stages of an important livestock pest, B. microplus. This is an important contribution to our knowledge and understanding of these molecules. However, a number of mis-statements are made which are relevant to this understanding and need to be rectified.  It has been shown that miRNA seed regions can have perfect Watson-Crick complementarity to the 3’UTR of mRNA (https://www.ncbi.nlm.nih.gov/pmc/articles/PMC5187787/#B53-ijms-17-01987). Because this is a well conserved mechanism, this can be used across multiple species. Could the investigators use this type of software as a cross validation for the mRNA-miRNA target predictions? During the mapping and novel miRNA identification, only the genome of I. scapularis was used. Could the investigators map the clean reads to other genomes to confirm the identified miRNA were truly novel? The miRNAs identified may be conserved within other Metastriata. Other specific comments that need to be addressed.

1.       At several places in the manuscript the authors state they are referring to “all stages.” This was not the case. Ticks have four developmental stages including the nymphs, which were not included in this study. In fact it is obvious from the methods section that the microRNA’s were extracted from only the off-host, non-feeding stages: the eggs, the preattachment larvae, and post feeding, engorged adults.  This is greatly relevant to the upregulation and down regulation of the microRNA’s. It is important that this distinction be made clear in the introduction and discussion.

2.       At line 151 in the text the authors use the designation “1” for miRNA’s that are not found in larvae or adults. Is it not simpler to say these were miRNA’s found in the eggs? The authors do not succinctly state what the designations “2” and “3” are for, leaving the reader to guess that these are found in larvae and adults.  Please define these designations clearly.

3.       Lines 45-46: these statements about B. microplus are all wrong. B. microplus is an obligate ectoparasite of bovines. They do always complete their development on cattle, and always on a single individual bovine. B. microplus are one-host ticks which make them unique and unlike 99% of other tick species which are almost all three-host ticks. Please remove these mis-statements.

4.       Title: following the scientific name Boophilus microplus add: (Acari: Ixodidae).

5.       Intro: Line 40, at first use, write out full scientific name: Rhipicephalus (Boophilus) microplus (Canestrini).

6.       Line 43: please add: Anaplasma

7.       Line 44: say “many”  not “most”

8.       Line 50, misspelling. Say “play” not “plat”

9.       Line 53, “movement”, do you mean “success”

Author Response

Point 1: The study provides statistics on the numbers, frequency, activity and function of miRNA’s in the developmental stages of an important livestock pest, B. microplus. This is an important contribution to our knowledge and understanding of these molecules. However, a number of mis-statements are made which are relevant to this understanding and need to be rectified.  It has been shown that miRNA seed regions can have perfect Watson-Crick complementarity to the 3’UTR of mRNA (https://www.ncbi.nlm.nih.gov/pmc/articles/PMC5187787/#B53-ijms-17-01987). Because this is a well conserved mechanism, this can be used across multiple species. Could the investigators use this type of software as a cross validation for the mRNA-miRNA target predictions? During the mapping and novel miRNA identification, only the genome of I. scapularis was used. Could the investigators map the clean reads to other genomes to confirm the identified miRNA were truly novel? The miRNAs identified may be conserved within other Metastriata. Other specific comments that need to be addressed.

Response 1: In general, when studying the function of miRNA, miReap and TargetScan software were first used to predict possible target genes of the target miRNA and screen the best matching relationship. The interaction relationship of dual luciferase reporter assays was used. However, in this study, we did not carry out this work for miRNA. Only the target prediction software was used to analyse possible target genes of potential novel miRNAs, and the functions of these target genes were analysed using GO and KEGG. The findings lay a foundation for future research on the regulatory function of miRNAs.

To identify the type and number of miRNAs in the sequencing data, this study mapped clean reads to all species, and differential expression of miRNAs was analysed between different samples. These unannotated small molecular sequences were used for prediction of novel miRNAs by miReap: http://sourceforge.net/projects/mireap/

Point 2:At several places in the manuscript the authors state they are referring to “all stages.” This was not the case. Ticks have four developmental stages including the nymphs, which were not included in this study. In fact it is obvious from the methods section that the microRNA’s were extracted from only the off-host, non-feeding stages: the eggs, the preattachment larvae, and post feeding, engorged adults.  This is greatly relevant to the upregulation and down regulation of the microRNA’s. It is important that this distinction be made clear in the introduction and discussion.

Response 2: R. microplus can complete its full lifecycle on cattle (one-host tick), so the ticks do not have a nymph stage on cattle. The developmental features have been described in the introduction and discussion sections.

Point 3:At line 151 in the text the authors use the designation “1” for miRNA’s that are not found in larvae or adults. Is it not simpler to say these were miRNA’s found in the eggs? The authors do not succinctly state what the designations “2” and “3” are for, leaving the reader to guess that these are found in larvae and adults.  Please define these designations clearly.

Response 3:  Thank you. This mistake has been corrected. "Conversely, the electron carrier and translation regulator protein showed a lower percentage in larvae than in eggs or adults. These data are provided in Figure 6 and annotated with “2” and “3”."

Point 4:Lines 45-46: these statements about B. microplus are all wrong. B. microplus is an obligate ectoparasite of bovines. They do always complete their development on cattle, and always on a single individual bovine. B. microplus are one-host ticks which make them unique and unlike 99% of other tick species which are almost all three-host ticks. Please remove these mis-statements.

Response 4: Thank you very much for the suggestion. This part has been revised.

Point 5:Title: following the scientific name Boophilus microplus add: (Acari: Ixodidae).

    Response 5: Thank you very much; (Acari: Ixodidae) has been added.

Point 6:Intro: Line 40, at first use, write out full scientific name: Rhipicephalus (Boophilus) microplus (Canestrini).

Response 6: Thank you very much. The sentence has been revised.

  Point 7:Line 43: please add: Anaplasma

Response 7: Thank you very much. The species has been added.

Point 8: Line 44: say “many”  not “most”

Response 8: Thank you very much. The word has been revised.

Point 9: Line 50, misspelling. Say “play” not “plat”

Response 9: Thank you very much. The sentence has been revised.

Point 10:Line 53, “movement”, do you mean “success”

Response 10: Thank you very much. The sentence has been revised.

Round 2

Reviewer 1 Report

The authors addressed all comments in a satisfactory manner. I am happy to recommend acceptance.

Author Response

Dear Editor,

Thank you very much for the comment, we will revise all suggestion or comment follow:

Results

Lines 159-183: This section is not clear. From methods, it seems this section should contain the description of genes targeted by the miRNA reported in this study (novel microRNAs and conserved known microRNAs). However, the authors begin by describing 'enriched genes' without mentioning explicitly that these are 'enriched target genes'. Please, reword for clarity.

R: Thank you very much for the suggestion, and the name of target genes have been added for clarity. For examples, extracellular region, macromolecular complex, membrane-enclosed iumen, etc were added to cellular components. PROK-G, ribozyme, ester hydrolase were added to molecular function. And the vesicle transport protein, ecdysone, heat shock protein have been added to biological processes category.

Materials and Methods

- Lines 535-536: At least as included in the document (lines 357-359), 'Figure 8' includes only a formula, not a plotting of 'log2-ratio and/or 'scatter plot'. Figures cannot be used to refer to something they do not display. Also, formulas do not need to be presented as figures, but immediatly after a text. For example, '...as calculated with the following formula:' (same applies to Figure 9) -but see the following point in regarding the formulas-.

R: Thank you for the suggestion, the Figure 8 and Figure 9 have been removed as Figure. And the formula have been added to document (lines 488-498).

- Concerning the formulas included by the authors as figures 8 and 9. Were those formulas developed by the authors for this study? If not, instead of adding the formulas, provide the reference of the manuscript and/or site from where the formulas were taken. If the formulas were developed for by the authors for this study, a formal statistical validation of the performance of each formula should be presented and included, at least as supplementary material.

R: Thank you for the comment. The formulas were reference from BGI Co. And the segment has been revised.

- The section 'Target prediction for novel miRNA candidates' (line 523) does not mentioned explicitly the methodology used by the authors to predict target genes. Please, be specific AND explicit on that. The description should allow others perform the analysis using the same methodology. There is also ambiguity in the description of this methodology. For example, it is mentioned (lines 529-530) that 'a combination of a dual luciferase reporting system and biological software is often applied', but results of 'dual luciferase reporting system' are not presented in this manuscript. Did the authors use 'a combination of a dual luciferase reporting system and biological software' to predict target genes? If not, remove this statement from the methods section. If yes, add the results of the combination of these methods. From what is written in results, it seems the authors used only the 'MireapV0.2 software' (line 176).

R: Thank you very much for your comment and suggestion, the segments have been revised in the section 'Target prediction for novel miRNA candidates'

Figures require major revisions. Below, the specific points that should be addressed.

General
- Letters naming panels (a, b, c and so on) should be placed consistently in the same position in relation to the panel images. For example, in the current version, letters are placed below the images (e.g., Figure 1) or in the top left (e.g., Figure 4). This has to follow journal specifications and be consistent in the whole manuscript.
- Remove unnecessary 'white space' between panels in the same figure.
- User mate colors and avoid shiny colors.
- In this manuscript, not all figure legends include the name of the figure (e.g., Figure 6). Each figure legend should include the name of the figure (i.e., brief sentence summarizing the content of the figure).
- Add figure legends below the figure, not above.

R: Thank you for your comment, All question have been revised, including Letters naming panels, white space, name of the figure and add figure legends below the figure. And the figure name font is bold.                 

Figure 1
- All circles in the Venn should have the same size. Numbers within circles should have the same font size.
- Font size of the legends indicating color codes should be the same. Only the first letter in the text of these legend should be capitalized. Spacing between words should be the same. Words should be fully spelled (e.g., 'Adu' should read 'adults').
- Remove 'Venn chart for uniq_sRNAs' from the top of the figure. In the figure legend that authors can mention that this is a Venn diagram.
- Use light colors to colored the Venn circles. Otherwise, the numbers are not clearly seem.

R: All the question have been revised. Including font size, color etc.

Figure 2
- Change the name of axis y to 'Frequency (%)'. Remove the name on the top of the image. The legend already provides this information.

R: Thank you for your suggestion,the name of axis y has been revised to 'Frequency (%)'. And the name on the top of the image has been removed.          

Figure 3
- Font sizes in this figure are too small. It is impossible to read a message in the figure. Resolution of the image is low. Please, improve.
- Remove the name on the top of the image. The legend should provide this information. The legend is to explain specifications of the figure in question. In this sense, it is not necessary to add a 'note' to the figure, and all necessary specifications can be part of the legend itself.

R: Thank you for your comment, they have been revised for font sizes in this figure. And the name on the top of the image has been removed, and the legend has been revised.  

Figure 4
- Font size of all axis y should be the same. Dedicate one histogram to each novel miRNA presented in the figure (as a result the figure will have 7 panels - and because each panel will now correspond to one miRNA it will not be necesary to add the code of the 'miRNA' inside the image, but only in the legend). This will allow the authors to homogenize the format and size of all bars. As the expression of each novel miRNA was measured in three stages, 'eggs', 'larvae' and 'adults', bars can be colored using a color per stage.
- If miRNA quantification included several biological and/or technical replicates (as mentioned in section 'Real-time quantitative PCR', line 551), these bars should include lines indicating mean values and standard deviation ranges.

R: Thank you for your suggestion, the figure has been redrawn, including font, color and description.

Figure 5
- Font size, and line thickness of all axis should be the same. Font size should allow the reader to see the names in axis. This is not the case in the current version of the manuscript.
- Remove brackets below histograms in panel 'b'. As the expression of each novel miRNA was measured in three stages, 'eggs', 'larvae' and 'adults', bars can be colored using a color per stage (if these colors match those use in previous figures to denote the same stage, it would be even better).
- If 'Panel “a” represents the expression level across the three libraries, which was detected by RT‒qPCR', this should be presented as expression levels (as in Figure 4) - not as 'Ct values'.
- Use one histogram per stage, and do not merged as in panel 'b'. Order histograms from eggs, to larvae to adults, as in Figure 4.
R: Thank you for your suggestion, the figure has been redrawn, including font, color and description.

Figure 6
- I understand that this type of results is usually presented this way. However, in this particular case, the information of the three life stages make dificult grasping the message of the figure. To solve this problem,I recommend splitting the figure in three panels. One panel for each Gene Ontology (GO) term used ('Cellular component', 'Molecular function' and 'Biological process').
- Not clear what numerals '1', '2' and '3' within the figure stand for. Please, explain in the legend or remove.
- Change 'Percent' to 'Percentage of genes in each GO category' in axis y.
- No need to add the same number in three different colors (axis y to the right). Please, use only one number in black color.

R: Thank you for your suggestion. In this segment, the Figure 6 was redrawn, and the  Gene Ontology (GO) term have been splitced the figure in three panels, and the numerals '1', '2' and '3' have been explained in the legend.
